# Monitoring Dewatering Fish Spawning Sites in the Reservoir of a Large Hydropower Plant in a Lowland Country Using Unmanned Aerial Vehicles

**DOI:** 10.3390/s23010303

**Published:** 2022-12-28

**Authors:** Linas Jurevičius, Petras Punys, Raimondas Šadzevičius, Egidijus Kasiulis

**Affiliations:** Department of Water Engineering, Vytautas Magnus University, 10 Universiteto Str., Akademija, 53361 Kaunas, Lithuania

**Keywords:** remote sensing, hydropower, fish spawning sites, large reservoir, dewatered areas, drawdown operations

## Abstract

This paper presents research concerning dewatered areas in the littoral zones of the Kaunas hydropower plant (HPP) reservoir in Lithuania. It is a multipurpose reservoir that is primarily used by two large hydropower plants for power generation. As a result of the peaking operation regime of the Kaunas HPP, the large quantity of water that is subtracted and released into the reservoir by the Kruonis pumped storage hydropower plant (PSP), and the reservoir morphology, i.e., the shallow, gently sloping littoral zone, significant dewatered areas can appear during drawdown operations. This is especially dangerous during the fish spawning period. Therefore, reservoir operation rules are in force that limit the operation of HPPs and secure other reservoir stakeholder needs. There is a lack of knowledge concerning fish spawning locations, how they change, and what areas are dewatered at different stages of HPP operation. This knowledge is crucial for decision-making and efficient reservoir storage management in order to simultaneously increase power generation and protect the environment. Current assessments of the spawning sites are mostly based on studies that were carried out in the 1990s. Surveying fish spawning sites is typically a difficult task that is usually carried out by performing manual bathymetric measurements due to the limitations of sonar in such conditions. A detailed survey of a small (approximately 5 ha) area containing several potential spawning sites was carried out using Unmanned Aerial Vehicles (UAV) equipped with multispectral and conventional RGB cameras. The captured images were processed using photogrammetry and analyzed using various techniques, including machine learning. In order to highlight water and track changes, various indices were calculated and assessed, such as the Normalized Difference Water Index (NDWI), Normalized Difference Vegetation Index (NDVI), Visible Atmospherically Resistant Index (VARI), and Normalized Green-Red Difference Index (NGRDI). High-resolution multispectral images were used to analyze the spectral footprint of aquatic macrophytes, and the possibility of using the results of this study to identify and map potential spawning sites over the entire reservoir (approximately 63.5 km^2^) was evaluated. The aim of the study was to investigate and implement modern surveying techniques to improve usage of reservoir storage during hydropower plant drawdown operations. The experimental results show that thresholding of the NGRDI and supervised classification of the NDWI were the best-performing methods for the shoreline detection in the fish spawning sites.

## 1. Introduction

In a previous study, water level (WL) fluctuations in the Kaunas hydropower plant (HPP) reservoir were analyzed [1]. It was found that the operations of the two large HPPs cause frequent short-term water level fluctuations. Daily drawdowns (the difference between min and max WL) range up to 0.4 m/day. These water level fluctuations frequently result in dewatered areas in the littoral zone of the reservoir. To observe and analyze these dewatered areas, a bathymetric survey was carried out.

The Kaunas HPP reservoir is situated on the East European Plain. The littoral zone (the nearshore area) of the reservoir is generally shallow with gentle slopes. It is a multipurpose reservoir, which is mainly used for power generation by the Kaunas hydropower plant and Kruonis pumped storage hydropower plant (PSP); however, it is also used for recreation, navigation, irrigation, industrial water supply, flood management, and recreational fishing [2]. Frequent water level drawdowns have a negative impact on the environment, which is discussed in many studies. Aquatic ecosystems in the reservoirs are sensitive even to small water level changes. Scientists claim that when water levels fluctuate in reservoirs with shallow and gently sloping littoral slopes, they experience a greater magnitude of water quality change than those with steep slopes [3]. In a review of the ecological impacts of winter water level drawdowns on lake littoral zones, many different ecological issues were discussed [4]. Water level fluctuations can have adverse effects on the environment, most notably on hydrologic and biotic processes, ranging in magnitude from the microscale to landscape level. Dewatered areas that are exposed during drawdown operations are susceptible to sediment desiccation and erosion from precipitation, and wind/wave action consolidating the exposed sediment layer [5,6], which increases the sediment bulk density. A study of Newnans Lake, a eutrophic lake in Florida, showed that short-term drawdowns can greatly accelerate erosion of the fine littoral substrate [7]. A relatively low rate of water drawdown and refill may enhance erosion by increasing the exposure time to wind/wave energy [8]. In reservoirs, these processes typically create barren shorelines with low habitat diversity and low species richness. Furthermore, they are likely to accelerate eutrophication processes and increase the risk of cyanobacteria blooms [9].

The littoral zone provides a spawning habitat for fish and water birds [10]. It is a physically complex system containing macrophytes, coarse woody debris, and other refuge for fish, which mediates competition and predation [11,12]. Although natural water level fluctuations are necessary for the aquatic ecosystem structure and function, water level regulation that exceeds natural variability may be very harmful to ecosystems in reservoirs [10,13]. Water level regulation and related habitat losses due to dewatering threaten ecosystem functioning and biodiversity in reservoirs [14]. Water level drawdowns tend to reduce benthic invertebrate density in the dewatered zones [15] as invertebrates are not particularly mobile, and they often become stranded and die from asphyxiation and desiccation [16]. All these factors tend to influence the abundance, density, and diversity of fish. Decreased fish growth rate, biomass, and abundance correlate with the losses of littoral physical habitat complexity [17]. Water level drawdown is a threat to fish species that use the littoral zone for all or part of their lives, especially in the spawning season [13,18]

The most abundant fish species in the Kaunas HPP reservoir belong to the phytophilic ecological group of fishes. It was determined that the main fish species whose spawning conditions may be affected by the fluctuating water level in the Kaunas HPP reservoir are pike, common roach, perch, and bream [19]. They usually spawn in the shallow areas of the reservoir, which contain aquatic macrophytes (where the water depth ranges from 0.2 to 2 m). Regulated water level fluctuations (e.g., rises and recessions) during spawning can negatively affect juvenile fish densities [20]. The Kaunas HPP reservoir operation rules are in force to mitigate the impact on the ecosystems and secure other reservoir stakeholders’ needs [2]. One of the most important issues is the assessment of dewater areas in fish spawning sites during the spawning period. Despite the wide scope of this issue, very few studies provide quantitative evaluations of the dewatering areas during short-term WL drawdown operations, which occur in hydropower plant reservoirs with shallow nearshore zones. Knowledge of potential fish spawning site locations, how they change in space and time, and how water level fluctuations affect the area is crucial for the optimal usage of reservoirs in terms of simultaneously increasing power generation and protecting the environment where it is most vulnerable. In previous research, a detailed traditional bathymetric survey of an area containing several potential spawning grounds (about 5 ha) was carried out. The water depth was measured with a water measuring pole with markings every centimeter and a plate on the bottom to prevent it from sinking into the sediment. A special iron weight attached to a graduated lead-line was also used to verify the results. The exact position of each measurement was collected with the Trimble R6 GNSS receiver. The point positioning accuracy using LitPOS RTK was 3–5 cm in the horizontal axis and 5–8 cm in the vertical axis. In total, over 1000 measurements were taken, elevation profiles were drawn, and a digital elevation model (DEM) was generated using the Kriging interpolation method, which was determined as the most suiting after statistical analysis of the accuracy of different interpolation methods. DEM was used for shoreline derivation and evaluation of the dewatering area in the selected fish spawning sites during different stages of HPP operation [1]. Traditional bathymetric surveys can provide reliable data concerning the selected area, but their application is limited as this kind of survey is challenging in difficult conditions.

The main focus of this study was to implement remote sensing (RS) techniques to track changes in drawdown areas in potential fish spawning sites. With the fast development of RS technologies, software, and the popularity of drones, surveying difficult areas has become easier and more accessible. Surveying with drones is a cost-effective and time-efficient method to collect data from the air, providing imagery or point cloud data from which a variety of deliverables can be extracted. According to various sources, data can generally be captured five times faster with drones than with traditional land-based methods [21,22]. For these reasons, several missions were carried out using Unmanned Aerial Vehicles (UAVs) to collect data from the aforementioned selected potential fish spawning sites at various water levels in the reservoir. After data collection, processing, and the analysis steps, the results were compared with the data gathered in the traditional field survey from the previous study.

Studying reservoir drawdown areas is a complex task, and several different approaches can be implemented. Different drone surveying methods for high-resolution river landscape mapping of the Belá River in the northern part of Slovakia were discussed in [23]. Three imaging methods for 3D model creation of the study area were used: (i) nadir, (ii) oblique, and (iii) horizontal. This minimized geometric error and captured topography under the treetop cover and overhanging banks. The article by [24] assesses the accuracy of UAV data processing using different software applications (Microsoft Photosynth, Agisoft PhotoScan and ARC3d) and discusses different processing schemes and validation strategies.

After data acquisition, processing, and validation, the resulting orthomosaic images can be used to detect the reservoir shorelines. A comparison of the shorelines detected at different stages of the HPP operation indicate the dewatered areas. There are many studies using various remote sensing techniques for shoreline detection. Moreover, various techniques have been used for different RS data sources. Satellite images (Sentinel, Landsat, etc.), data collected by Light Detection and Ranging (LiDAR), and aerial photography from the planes or UAVs were used in previous studies, some of which are discussed in Section 4. In a review of shoreline detection using optical RS carried out by [25], different water segmentation techniques were discussed and some were implemented in this study.

Spectral reflections from different surfaces are the key to the techniques used in this study. In 1996, McFeeters introduced the Normalized Difference Water Index (NDWI), which makes use of reflected near-infrared (NIR) and visible green light radiation for assessing quantity (e.g., the surface area) and quality (e.g., the turbidity) of water resources [26]. Hanqiu Xu modified the NDWI by substituting middle-infrared for near-infrared radiation in the NDWI in order to enhance open water features while efficiently suppressing and even removing built-up land noise and soil noise [27]. Indices can be used to obtain a simplified data representation by separating and highlighting different surfaces from one another. Another commonly used index is the Normalized Difference Vegetation Index (NDVI). The NDVI is calculated by measuring the difference between near-infrared (which vegetation strongly reflects) and red light (which vegetation absorbs). It is able to quickly delineate vegetation and vegetative stress, which has many uses in commercial agriculture and land-use studies [28]. In our case, this was also useful for indexing water, because infrared radiation is strongly absorbed by water and thus it can easily be identified by the low (close to zero or negative) NDVI values. Moreover, this index helps to separate aquatic and land vegetation.

Part of the study was carried out with a drone equipped with a simple RGB camera. Data of NIR were unavailable for these datasets; the NDVI and NDWI could not be calculated. Different combinations of red, green, and blue bands were used to separate water, aquatic, and land vegetation. The purpose was to establish the best-performing technique as compared to methods that rely on NIR. In this way, the technique could be adopted for datasets that only contain red, green, and blue bands, which correspond to the visible part of the electromagnetic spectrum and can be captured with a simple camera sensor. The Visible Atmospherically Resistant Index (VARI) and the Normalized Green–Red Difference Index (NGRDI) indices using only the visible range of the spectrum were calculated. The VARI was proposed by [29] for remote vegetation fraction estimation. The index was found to be minimally sensitive to atmospheric effects, allowing for vegetation fraction estimation with an error of <10% in a wide range of atmospheric optical thicknesses. In addition, the NGRDI was used in comparison with the NDVI for highlighting water. The NGRDI is similar to the NDVI, but uses green instead of NIR bands [30]. 

This research differs from others in the following ways: We are interested in changes in the submerged area in potential fish spawning sites, which means that the shoreline must be detected in areas containing aquatic macrophytes. In areas that are clear of aquatic vegetation, many known techniques, as discussed above, can be implemented; however, in the areas with thick water vegetation, it is very difficult to detect and segment water. Therefore, the best methods and indices to achieve this were evaluated.

A spectral signature analysis was carried out for aquatic and land vegetation in order to separate one from the other. This means that potential fish spawning sites could be detected over the entire reservoir and their area could be calculated. This knowledge would allow us to locate the most vulnerable areas and estimate how much water level fluctuations affect them. A similar study of spectral signatures was conducted for automatic blackgrass weed mapping using a supervised classification technique [31]. A number of advanced techniques were used: feature generation to enhance the feature discrimination ability, feature selection for dimension reduction, Random Forest (RF) for classification, and guided filter for spatial information enhancement.

The aim of the study was to implement modern remote sensing techniques to investigate dewatering areas in the fish spawning sites. Current assessments of spawning sites in the reservoir are mostly based on studies that were carried out in the 1990s. Surveying fish spawning sites is typically a difficult task that is usually carried out by performing manual bathymetric measurements due to the limitations of sonar. Majority of the devices based on echo-sounding (sonar) cannot operate in the shallow waters containing aquatic macrophytes—the typical spawning sites for many species of fish. Our hypothesis is that RS can be used to assist in surveying difficult and vulnerable areas, where the current and accurate data are needed. This knowledge could assist in making decisions for better use of reservoir storage while increasing power generation.

Various tasks were set for this study:To review current practice related to reservoir drawdown/hydropower operating rules and policies;To conduct surveys of the study area using remote sensing methods;To detect shorelines during different stages of reservoir operation;To evaluate the effectiveness of different methods of shoreline detection in various nearshore conditions;To analyze the spectral signatures of aquatic macrophytes and to evaluate the possibility of implementing the results for automatic classification over the entire Kaunas HPP reservoir;To summarize the findings in order to provide insights for deriving the optimal operating rules for the reservoir.

## 2. Materials and Methods

### 2.1. Object of Study

The Kaunas HPP reservoir is the largest artificial water body in Lithuania. The reservoir was created by damming the Nemunas River in 1959. The reservoir is used by two large hydropower plants—the Kaunas Hydropower plant, which has an installed capacity of 101 MW and a rated head of 20.1 m, and the Kruonis Pumped Storage Hydropower Plant (PSP), which currently has an installed capacity of 900 MW from four turbines and a rated head of 93.6–111.5 m (depending on the water level in the upper reservoir). The Kaunas HPP reservoir is mainly used for power generation but is also used for recreation, navigation, irrigation, industrial water supply, flood management, and recreational fishing. Water level fluctuations in the Kaunas HPP reservoir are mostly dependent on power plant operation and the upstream inflow from the Nemunas River. The impounded area of the Kaunas HPP reservoir at a normal water level (NWL) is 63.5 km^2^ and the volume is 0.46 km^3^. The effective capacity of the reservoir available for hydropower is 0.22 km^3^. The average depth of the Kaunas HPP reservoir is 7.3 m, and the depth of the reservoir is 10–12 m in the lower part and about 4–5 m in the upper part [32].

Operating two large HPPs causes water level fluctuations in the reservoir. The reservoir operation rules were established to protect the environment and ensure other stakeholders’ needs. There are two main operation regimes. During normal operation, the permitted water levels in the reservoir are between 43.5 and 44.4 m a.s.l. A daily water level drawdown of ±0.4 m (from the normal headwater level NWL) is allowed. During the fish spawning period, the operation of the Kaunas HPP and Kruonis PSP is restricted. The fish spawning period occurs between 1 April and 30 June. During the restricted regime, the headwater elevation of the Kaunas HPP reservoir must be between 43.7 and 44.0 m a.s.l. A maximum difference of 10 cm between the highest and the lowest daily water level (or, in other words, drawdown) is allowed. A daily water level change of 20 cm is allowed in the reservoir if, each year, the operator commissions an environmental study according to the Environmental research program and compensates for the environmental damage if it was caused [2].

The northeastern shore of the Kaunas HPP reservoir was selected as the study area. It is one of the main spawning sites for perch, common roach, and common bream [19], and is located in between the Kaunas HPP and Kruonis PSP. The size of the selected area is approximately 5 ha and contains several potential fish spawning sites. The bottom of the reservoir in this area is very shallow and gently sloping. A detailed survey of the area using traditional methods was conducted beforehand to gather data as a reference. The study area contains several different shore conditions—shallow sandy beaches, steep banks covered with land vegetation, and several areas containing aquatic macrophytes (reeds and bulrush) that are typical spawning and nursing grounds for the fish. It was important to select an area with different conditions that are common in different parts of the reservoir in order for the findings of this study to be relevant on a much larger scale. The map of the Kaunas HPP reservoir and study location is presented in Figure 1.

### 2.2. Shoreline Detection Using Remote Sensing Techniques

#### 2.2.1. Data Collection and Processing

Planned drone flights were carried out throughout the summer and autumn of 2021 during different water level fluctuations in the reservoir. There were two drones used for the study, i.e., DJI Martice 200 equipped with the 12 MP Sentera AGX710 multispectral camera and Autel EVO II Pro equipped with the conventional XT705 20 MP RGB camera. For each flight, the same mission was carried out in the same area (approximately 5 ha). The survey area, flight paths, and settings were prerecorded and used for each mission. During the mission, the drones took approximately 350 images of the area from a 60 m elevation with an 80% front and 75% side overlap. Drone flight mission information is presented in Table 1.

In order to consistently process the images with a high accuracy, 10 permanent ground control points (GCP) were established. The photogrammetric process needs the support of control points to be able to scale and to georeference the model [33]. Most of the GCPs were placed alongside the shoreline, where the accuracy is most crucial. The remaining GCPs were established at the start, middle, and end of study area for an overall better referencing performance. 

Coordinates of those points were measured with the Trimble R6 GNSS receiver. GCPs were visible in the photos that were taken and referenced to the exact coordinates with the photogrammetry software. Water levels were measured at the start and end of each mission using a water level gauge, which was leveled with the Trimble R6 GNSS receiver to determine the accurate water level altitude during each mission.

Data from UAVs were processed using photogrammetry programs, i.e., Agisoft Metashape Pro and Pix4Dmapper. Drone surveying and mapping is commonly used in civil engineering, agriculture, land surveying, and other fields. The workflow of drone surveying and data processing with different applications is discussed in numerous articles, some of which [21,22,23,24] were described in Section 1.

The techniques discussed in the articles and software documentation [34] were used to process the raw data collected with the UAVs. Images were referenced and aligned, and images with visible GCPs were tied to the exact coordinates. Then, camera calibration was performed. After the alignment, tie points of the images and dense point clouds were generated using high-accuracy and moderate-depth filtering, which performed the best. Additional filtering was performed according to the dense point cloud confidence readings. After the correction of the dense point clouds, digital elevation models were created and orthomosaics were built according to the DEMs. The resulting orthomosaics were the final products and were used to compare the fish spawning sites at various water levels in the reservoir. The resulting orthomosaics were of a very high resolution, with a ground sample distance (GSD) of approximately 1.5 cm. The aligned GCP’s precision measured as the average rate of root mean square errors (RMSE) of all GCPs after alignment was 0.032 m on average.

#### 2.2.2. Data Analysis and Shoreline Detection Techniques

Various indices were calculated using the spectral bands of the orthomosaics in order to separate water from other areas (sand, land vegetation, etc.). Indices were calculated to simplify complex data amalgamations, e.g., orthomosaic images. The NDWI and NDVI were used to highlight the water from other surfaces during the analysis of multispectral images. These indices were derived from the NIR, Green (G) and Red (R) spectral bands. They are dimensionless indices that describe the difference between visible and near-infrared reflectance of surfaces. The orthomosaics created from RGB images did not have an NIR band available. For this reason, the VARI and NGRDI indices were used in the same application to highlight the pixels that represented water. Equations of the aforementioned indices can be found in the literature that was reviewed in Section 1.

The resulting raster images of various indices were thoroughly analyzed. The index values of four different surface conditions in the study area were compared. The thresholding technique was used to segment images into several classes—water, aquatic macrophytes, land vegetation, and sand. Generally, a threshold limits can be selected by trial and error (manually) or using the peak–valley method of histogram segmentation [35]. In this instance, index value limits were set manually for each surface. The resulting segmented images were used to detect the shorelines.

Two different areas were analyzed to evaluate the performance of different approaches. The first area was a wavy shoreline with steep banks. The water in that area was mostly free from aquatic vegetation, so the shoreline, between land vegetation that covers the banks and the water, was well-defined. The more challenging part was between the bends of the shore, which were shaded from the sun. The second area was the opposite—the banks were very shallow, covered with sand, and clear from vegetation. One part of the sandy beach was covered with silt. Thresholding and unsupervised classification were carried out for the raster images containing NGRDI, VARI, NDVI, and NDWI values.

### 2.3. Shoreline Detection in the Fish Spawning Sites

In the spawning sites, shoreline detection was complicated because the aquatic vegetation obstructed the water. For this reason, two different image classification techniques were used to obtain a simplified data representation of a set of homogeneous and natural regions (called classes) to help detect the shorelines. Unsupervised classification was carried out for all the raster images. The outcome was groups of pixels with common characteristics (similar index values or spectral reflectance) based on the software analysis of an image without the user providing sample classes. This task was carried out with ArcGIS PRO software using ISODATA (ISO) clustering. Unsupervised (calculated by the software) classification methods are explained in [36]. Hierarchical methods are divided into agglomerated and divisive; non-hierarchical algorithms are divided into density-based, partitioning, grid-based and others. Algorithms such as Birch and Cure are hierarchical; K-means, Fuzzy C-Means, and ISODATA are partitioning; DB-Scan and OPTICS are density-based. New methods—the weighted density-based optimized classification method and the automatic density-based optimized classification method—are proposed in [37].

Supervised classification was mainly used in an attempt to improve the results in the areas with dense aquatic vegetation. For this step, orthomosaics and raster images of the NDWI and NGRDI values were analyzed. These indices were chosen as they performed the best for multispectral and RGB image segmentation. Supervised classification was carried out by defining sample areas, otherwise known as Regions of Interest (ROI). The defined areas were analyzed using the supervised machine learning algorithm (Random Forest) to distinguish the characteristics of the provided areas. The RF ensemble classification and regression approach was developed by [38]. It exhibits outstanding performance and is widely used. The algorithm was used to segment the rest of the images into regions that can be associated with the spectral signature or index values of the provided sample class data. Four classes (different surfaces) were defined: water, aquatic macrophytes, land vegetation, and sand. The described surfaces were not very homogenous. For this reason, each class had several subclasses defining them. Although increasing the number of training polygons was not always the optimum solution, after adding each subclass, the results were checked by classifying several small areas throughout the dataset. After the best classification performance was reached, the same training areas were used to classify the entire raster image containing NDWI and NGRDI values. The same procedure was repeated for all the datasets from all flights. This technique allowed us to fine-tune the segmentation of the surfaces, but it required more time to define the classes and train the algorithm. Classified images were used to derive the shoreline in the potential fish spawning sites.

Various water body segmentation techniques by satellite image classification were discussed in [39,40,41]. The assessment of accuracy and quality of image classification was carried according to the forementioned studies by calculating Overall Accuracy (OA) and Kappa Coefficients (KC). Accuracy assessment points were generated across the segmented raster images. Accuracy assessment points were randomly distributed within each class, where each class has a number of points proportional to its relative area. In total, 500 points were created for each dataset containing assigned class values during classification and ground truth values. Ground truth references were delineated manually by careful visual interpretation of the very high-resolution the orthomosaic images. Areas containing aquatic macrophytes were left out from this process as they are very heterogeneous and determining ground truth values would be very difficult. Confusion matrices were computed, and OA and KC values were calculated [40]. These coefficients showcased the classification ability to separate water from the vegetation. Classification accuracy in the areas containing aquatic macrophytes was evaluated by careful visual inspection as there are too many uncertainties in such areas to assess by OA and KC.

Raster images containing NDVI and VARI indices were not segmented using supervised classification as these indices did not improve the results in the first part of the study (Section 2.2.2). The results of traditional surveying and remote sensing were compared. The shorelines were determined from the DEMs that were generated from data collected in the traditional bathymetric survey. The methodology and the short description of the study are provided in Section 1. A flowchart of the UAV data collection and shoreline detection in the potential fish spawning sites is presented in Figure 2.

### 2.4. Spectral Analysis of Aquatic Vegetation

A DJI Matrice 200 equipped with multispectral camera was used on 8 June 2021. The water level in the reservoir during the flight was 43.76 m a.s.l. The multispectral camera captured images with six bands (wavelengths range was 400–900 nm): red, blue, green, red edge 1&2, and near-infrared. These data allowed for a deeper analysis of the area. The main purpose was to study the spectral footprint of the aquatic macrophytes and land vegetation. Spectral reflections of four different surfaces were analyzed: water, land vegetation, water vegetation, and bare sand. Areas that represented studied areas were carefully selected and their spectral signatures were calculated using QGIS. The Bray–Curtis similarity index was calculated to estimate if the surfaces could be separated [42].

## 3. Results

### 3.1. Shoreline Detection Using Remote Sensing Techniques

The different shoreline detection techniques exhibited different strengths and shortcomings when appd in different conditions. A comparison of the resulting shorelines is presented in Figure 3.

The first set of pictures was the orthomosaics of the two selected locations (a and b). The shorelines were detected by visual inspection from high-resolution orthomosaic images and are shown as the red line. This remains in all the other pictures as a reference. The second set of pictures is the classified NGRDI pictures using unsupervised classification. The algorithm detected five different classes, which is the same outcome as that of the VARI raster image classification. Overall, the outcome of the NGRDI and VARI unsupervised classification was essentially the same, and NGRDI was chosen for further investigations as it is easily readable, where values range from −1 to 1. The outcome of the classified NGRDI/VARI raster images was the worst. There was a great deal of noise (incorrectly classified pixels) in the water and land areas. In the first location (a), the shoreline was the least defined, and in the second location (b), the algorithm was unable to separate water from the wet sand on the shore. The third set of the pictures is classified images of NDVI and NDWI values. In both cases, the unsupervised classification algorithm (ISO clustering) detected seven classes and both outcomes were very similar. The NDWI represented water with a little less noise as it is designed to do, but overall, in this case, the two techniques were essentially interchangeable. The use of the NIR band helped to separate water from the vegetation. In the first location, the shoreline was best defined and the shadows did not have an impact on the outcome. However, in the second location, this method was unable to separate the water from the sand. Bare wet sand absorbs NIR in a similar manner to water. In both cases, the difference in the indexed values was not enough to separate wet sand from water. The fourth set of pictures was classified orthomosaic. The algorithm detected five different classes. Unlike in the other cases, it separated a class for shadows (dark areas). The noise in the classified pictures was average. The algorithm was considering the entire light reflectance spectrum captured by the camera and was able to separate water from sand, assigning it a different class. However, this was not done accurately, i.e., the area with silt on the sand was assigned to the water class and the shallow part of the reservoir with sand on the bottom was assigned to sand. The resulting shoreline in the shallow part of the reservoir was not accurate.

The assessment of accuracy and quality of unsupervised image classification was carried out to determine the best-suiting indices for highlighting water. The results of accuracy evaluation are provided in Table 2.

The accuracy assessment showcased that the best index for water identification was NDWI, which was to be expected. NGRDI showed to be a valuable option of water and vegetation identification if NIR is not available. It has performed well at OA of 92% and KC 0.837, which is according to categorization of the Kappa statistic—almost perfect [43]. These indices showcase classification ability to separate vegetation from the water area. It is the most important part in our study as in most cases shoreline marks the transition between land vegetation and the water area. But there are other surfaces that are difficult to classify—sand (wet and dry) and areas containing aquatic macrophytes. These surfaces are too heterogeneous to calculate OA and KC.

The accuracy assessment of the classification of the forementioned surfaces was done by careful visual inspection. To improve the classification accuracy, the thresholding technique was applied for the NGRDI, NDWI, and NDVI raster images. This technique allowed us to assign an interval of index values to different classes. This technique allowed us to fine-tune the results, to study the index values of certain areas, and to assign their range to a different class. Thresholding the NDVI and NDWI allowed us to separate sand, but the part with the silt on the sand had no distinguishable difference from the water. Thresholding the NGRDI was promising. The index is calculated from the reflections of the wavelengths in the red and green spectrum and was able to separate sand from water relatively well (see the fifth set of images in Figure 3). The dark areas in the images were the weak points, with the very dark and very bright areas having similar index values. In some cases, this can denote dry areas, such as sand, and other times shady areas, which could also be water, and both areas will have a very similar index value. Such areas must be doublechecked and conscious decisions must be made. Overall classification of raster images with NDWI values can be used to automatically detect shorelines, but it would not work accurately if the shores were very shallow and sandy. Other indices and methods are just a tool to help highlight water from other areas, but the shorelines should be defined manually. The NIR spectrum gives more information about the surfaces and is not affected by shadows to the same extent. The results indicate that orthomosaics containing the visual RGB spectrum can also be used for shoreline detection coupled with the thresholding technique.

Performing supervised classification using the machine learning algorithm slightly improved the results of image segmentation. The outcome was compared with the thresholding of NGRDI in the two areas with different shore conditions (Figure 4).

In the case of the shallow nearshore with sandy banks (a), the transitional area that is constantly affected by the waves (wet sand) was noisy, with some pixels being classified as water and others being classified as sand. The area with wet sand is marked between the two red lines. NGRDI thresholding is more useful in this scenario because the water, and wet and dry sand have some separation from each other, according to the NGRDI values. In the case of steep banks, where the shoreline can be defined as the separation between water and land vegetation (b), the supervised classification worked very well and the shoreline was defined very accurately. The resulting image of NGRDI thresholding has more noise, which would make automatic shoreline detection difficult without filtering. Overall, manually defining different classes for the machine learning algorithm improved the results and classified images had less noise. The separation of water from the sand was more pronounced and the algorithm was able recognize more surfaces. The resulting images were more detailed, but it struggled in the same conditions as unsupervised classification, i.e., shallow sandy areas where improvement was just slight, and trying to define those areas with more input for machine learning algorithm resulted in more noise in the rest of the image (pixels that were assigned to wrong classes), which produced the opposite effect. Classification of the images with NGRDI values did not produce better results than thresholding. Thus, supervised classification is not always the best solution, and sometimes, depending on the conditions of the nearshore of the reservoir, NGRDI thresholding is a better approach. Studying NGRDI values of different surfaces can result in accurate image segmentation into different classes, and in the case of the shallow sandy areas it was the most accurate option.

### 3.2. Shoreline Detection in the Fish Spawning Sites

It is possible to detect shorelines in areas with medium-density aquatic macrophytes (bulrushes in this case). NGRDI thresholding was able to indicate water in such conditions with relative ease (Figure 5).

Remote sensing techniques have great advantages over the traditional methods in these areas because the vegetation does not obstruct the water, and shorelines can be detected easily. Traditional bathymetric surveys in such areas are difficult. Echo-sounding is not possible due to vegetation, and measuring points with an RTK-GPS receiver is physically demanding. Areas in which traditional methods have a clear advantage are those with dense aquatic vegetation (Figure 6).

When the density of aquatic vegetation is high (in this case, areas with dense and high reeds), it is very difficult to detect the shoreline using RS techniques. Dense aquatic macrophytes almost completely obstruct the water. There were not many pixels indexed as water and judging the location of the shoreline was very difficult. The thresholding of NGRDI allowed for better separation of the surfaces, but in these cases, the shoreline could only be approximated.

The shorelines derived from bathymetric measurements (according to the generated DEM) and RS were compared (Figure 7).

It is noticeable that the shorelines derived from RS were more accurate in areas with an absence of or medium-density aquatic vegetation. The shorelines derived from the bathymetric measurements were not as detailed because the digital elevation model was interpolated between the measurement points. Generally, the results derived from traditional land-based surveys will never be as detailed because it is not possible to measure enough points to match the spatial resolution of the data collected by RS. Although bathymetric measurements in areas with dense aquatic vegetation are difficult, but it was possible to obtain accurate bathymetric data of the reservoir, which is where it has an advantage over remote sensing. The shorelines determined using the morphological method might not have been as detailed, but they were accurate enough to calculate dewatering areas in the fish spawning grounds at any given water level throughout the entire operating range of the HPPs, where the shoreline detection in such areas using RS is possible but complicated and might be not as accurate depending on the conditions.

### 3.3. Spectral Analysis of Aquatic Macrophytes

A spectral signature analysis was carried out for the orthomosaic image captured with the multispectral camera (Figure 8).

Judging from the graph, sand produced the highest reflection of blue, green, and red. This is expected because it is a light color, i.e., it reflects the light the best and the values in the range of NIR are lower, because NIR is absorbed by the sand. Water reflects light in the blue and green spectrums the most. In the summertime, water in the Kaunas HPP reservoir has a light green tint and the NIR spectrum reflectance is lowest out of all of the analyzed areas, because water absorbs the NIR the most. The reflectance in the near-infrared region of the electromagnetic spectrum suggests that the water may have contained a small amount of algae. The most important part of this analysis was the comparison of land and water vegetation. It is visible in the diagram that overall water vegetation had a higher reflectance in the RGB range, with the largest difference being noticeable in the blue spectrum. Some yellow reeds made up the water macrophytes; for this reason, red and blue reflectance was higher as compared to the land vegetation. The NIR reflectance was quite similar for land and water vegetation, with the land vegetation reflecting the NIR a little more, because it was denser in general. A statistical analysis of the spectral reflectance of land and water vegetation was performed for further investigation (Table 3).

The standard deviation of the reflection values in the visible region of the electromagnetic spectrum of the water vegetation class was higher because it comprised different types of aquatic macrophytes with different signatures. Analyzing the spectral signatures of the subclasses that were defined for land and water vegetation, the Bray–Curtis similarity index ranged from 71 to 89%.

## 4. Discussion

According to our previous study on dewatering areas during drawdown operations and water level fluctuations in the reservoir [1], we demonstrated that large multipurpose reservoirs in lowland countries are susceptible to various ecological problems owing to their morphology, which was discussed in the introduction. Accurate data of the most vulnerable littoral areas are crucial for the optimal and environmentally sound operation of the reservoir. Generally, shallow foreshore areas are very difficult to survey because they are too shallow for echo-sounding and are usually measured using traditional bathymetric surveying methods. Traditional types of surveying in reservoirs are difficult and labor intensive because foreshores are often hard to access due to aquatic vegetation and accumulated silt on the bottom. Remote sensing techniques showed to be a valuable addition to traditional methods. It is possible to obtain bathymetric data of shallow water bodies from multispectral or RGB imaging. These methods rely on the Beer–Lambert law, which describes the absorption effect as light passes through transparent media (water) [44]. This is the physical principle underlying the measurement of water depth from brightness levels in captured imagery. There are plentiful studies on the matter [45,46]. For example, after comprehensive calibration, the authors introduced an automated bathymetric mapping method capable of a 4 m^2^ spatial resolution with a precision of ±15 cm for remotely sensed datasets [47]. In the preliminary assessment of airborne hyperspectral coastal bathymetry capabilities based on International Hydrographic Organization (IHO) standards, the authors state that the hyperspectral bathymetry estimations were close to being consistent with an IHO order 2 standard up to a 14 m depth in the first test location and up to a 5 m depth in the second location [48]. The datasets were collected over Mayotte and the Geyser Bank, north of Madagascar, Indian Ocean. However, there is a weakness inherent in photogrammetric methods due to uneven light. It has been shown that the red color band has a greater sensitivity to depth than blue or green [46] but it does not penetrate the water column as deeply. These studies use depth and water color relationships that are site-specific and require ground surveys to calibrate this process [49], and the accuracy suffers when there are changes in the substrate material, overhanging vegetation, surface disturbance from waves and shadows, etc. [47]. 

An article by [50] presents a generic processing chain that covers all modules required for operational flood monitoring from multispectral satellite data. Segmentation of the water extent is performed by a convolutional neural network that has been trained on a global dataset of Landsat TM, ETM+, OLI, and Sentinel-2 images. In the article, various water segmentation techniques were discussed and the authors utilized an interesting approach that could be used for different forms of satellite data. This approach is appropriate for monitoring large floods, when the changes in the water surface area are large; however, for our study, the resolution of the satellite images was not sufficient to track relatively small changes. More detailed and accurate data are needed, and each dataset was analyzed individually to obtain the most accurate results possible.

Another approach involves the use of airborne Light Detection and Ranging (LiDAR) technology to conduct bathymetric surveys of shallow water regions. There are various bathymetric LiDAR systems. The majority make use of a green laser for the bathymetric survey. However, the availability of these devices is still relatively low, mostly due to their high prices. LiDAR is only feasible for relatively shallow and clear waters. During the study of two lakes in Poland, it was concluded that a LiDAR sensor can be used for measurements in the littoral zone (up to 1.6 m), and in deeper areas the accuracy significantly decreased [51]. In another study, an assessment of the ability to map river bathymetry using airborne LiDAR was carried out [52]. Data were collected over 220 river kilometers in the Yakima and Trinity River Basins in the USA. The mean error of water depths varied from 0.04 to 0.52 m. A multistep morphological technique that works by utilizing the digital elevation models (DEMs) obtained from LiDAR with respect to a tidal datum was discussed by [53]. It remains unclear whether LiDAR surveying is accurate in areas containing water macrophytes such as reeds and bulrush. For our purposes, it would be feasible to use a regular topographic LiDAR system when the water level in the reservoir is at its lowest point.

We took the shoreline detection approach for this study using photogrammetry. The drawback of this method is that it requires multiple surveys, but it was possible to determine the shoreline in areas containing aquatic macrophytes where other methods would fail. It is much easier to conduct RS surveys in difficult areas, and surveying is much quicker compared to traditional surveys. Furthermore, traditional surveys cannot match the resolution of the data collected by remote sensing (depending on the RS data source). There are many different techniques for shoreline detection, as was explored in the Methodology section.

Different water segmentation methods were implemented, including machine learning, to establish the optimum methods for shoreline detection in shallow nearshore areas with different conditions. A very interesting comparison and assessment of different object-based classifications using machine learning algorithms were presented in the article [54]. The study was implemented for bergamot and an onion crop located in Calabria (Italy). Four classification algorithms were assessed: K-Nearest Neighbor, Support Vector Machines, Random Forests, and Normal Bayes. The Remote Sensing and Geographical Information Systems software Library was used in the image segmentation step. A similar approach can be utilized for aquatic vegetation classification and segmentation, which could be the basis for further research.

Our study demonstrated that image classification and thresholding of various indices can be used to highlight pixels that represent water in areas containing aquatic macrophytes. It works well in areas with medium-density vegetation, but using these techniques in the areas covered with dense aquatic vegetation often does not clearly separate the shoreline. Shorelines can be approximated by judging the density of water pixels in the aquatic vegetation, but it is difficult and not as accurate as in areas that are clear from vegetation.

There is no universal method to detect shorelines using RS, i.e., in different conditions, different methods are better-suited. NGRDI thresholding showed the best results in areas with shallow and sandy shorelines, and supervised classification was typically the best method for separating water from vegetation. Unsupervised classification of NDWI was also accurate and performs relatively quickly if multispectral images are available. These findings make surveying more accessible with simpler equipment. Moreover, when assessing shorelines with washed-up silt, a visual analysis of orthomosaics was the most reliable method because the other techniques failed to separate these areas without introducing excessive noise to the segmented images. In difficult conditions with dense aquatic macrophytes, shoreline detection was very complicated. It was necessary to double check the results of all water surface identification techniques, because each of them had some percentage of incorrectly identified pixels (noise) and it was relatively easy to incorrectly identify the shoreline. In such conditions, traditional manual bathymetric surveying is the most reliable method, but it is very challenging.

## 5. Conclusions

The research area of approximately 5 ha is very small compared to the total area of the reservoir, which is 63.5 km^2^ at normal water level. Moreover, various spawning sites have different morphologies and nearshore conditions, so these results cannot be applied to the entire reservoir. The purpose of this research was to explore the possibility of utilizing alternative surveying methods and to compare the outcome with traditional morphological surveys. The successful implementation of modern surveying and analysis techniques could provide very useful insights to assist in improving reservoir storage usage while increasing power generation. 

After comprehensive data analysis, it was determined that NDWI was the best-suiting index for highlighting water from the vegetation (OA of unsupervised classification—97.4%, KC—0.947). NGRDI also performed very well for the orthomosaics that did not contain NIR. The OA of unsupervised classification of NGRDI was 92% and KC 0.837. These findings can be adopted for further research in areas with similar conditions that are common for the reservoirs in the lowland countries. The results of the spectral signature analysis suggest that image segmentation could be carried out using supervised classification. Medium resolution multispectral mosaics are available for most parts of Lithuania (GSD is approximately 0.2 m); therefore, future research will focus on using the findings of this study for the automatic detection and mapping of potential fish spawning sites over the entire reservoir and for assessing the degree of accuracy. This information would be invaluable, as current assessments of spawning sites in the reservoir are mostly based on studies that were carried out in the 1990s.

## Figures and Tables

**Figure 1 sensors-23-00303-f001:**
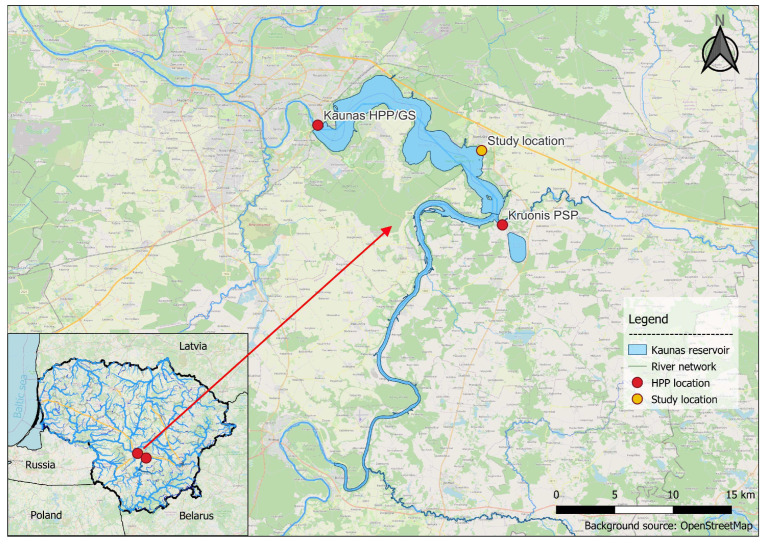
Map of the reservoir of the Kaunas Hydropower Plant.

**Figure 2 sensors-23-00303-f002:**
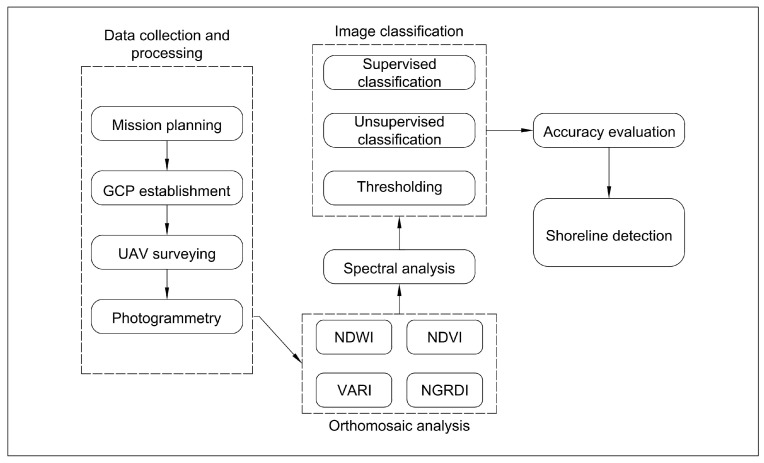
Flowchart of the UAV data collection and shoreline detection in the potential fish spawning sites of the Kaunas HPP reservoir.

**Figure 3 sensors-23-00303-f003:**
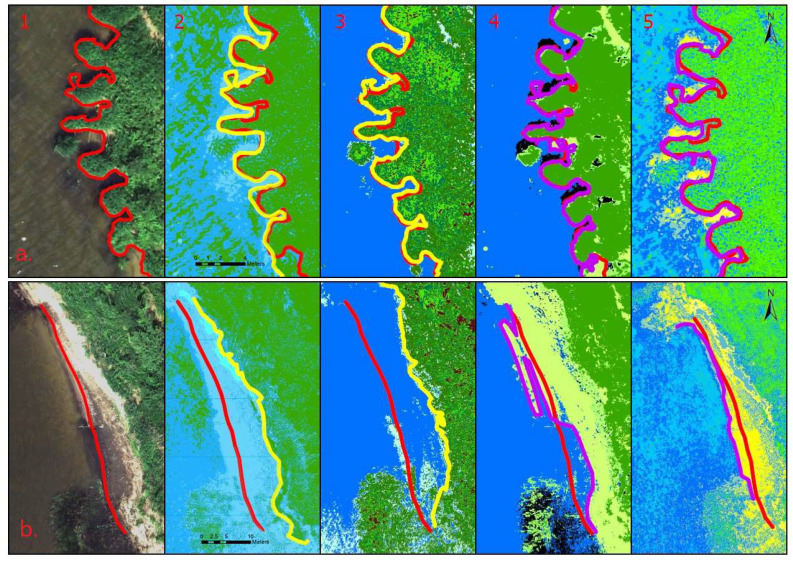
Comparison of the different shoreline detection techniques in different shore conditions: (**a**)—reservoir banks are steep with land vegetation covering them; (**b**)—very shallow sandy shores. 1—Orthomosaic of the area with the shoreline marked as a red line (it stays in the other pictures as a reference). 2—Classified raster containing NGRDI values with the shoreline marked as a yellow line. 3—Classified raster containing NDWI values with the shoreline marked as a yellow line. 4—Classified multispectral orthomosaic with the shoreline marked as a magenta line. 5—NGRDI thresholding with the shoreline marked as a magenta line.

**Figure 4 sensors-23-00303-f004:**
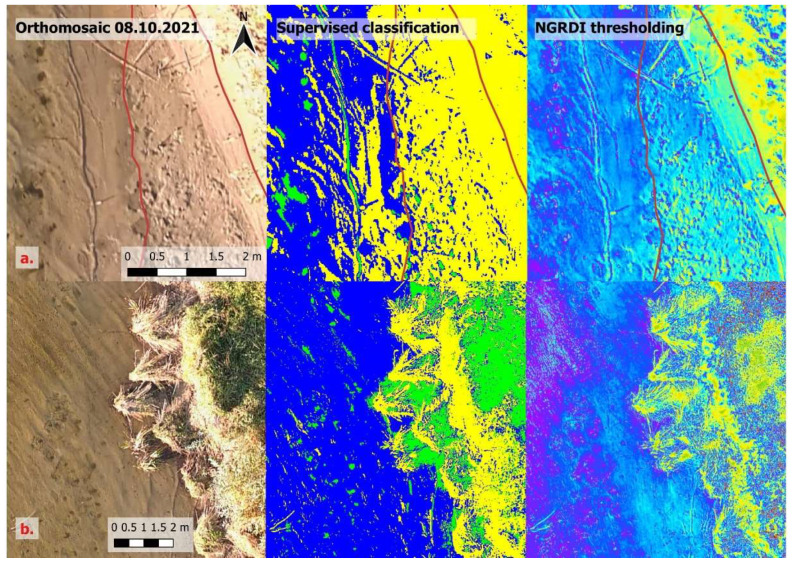
Shoreline detection using supervised classification of orthomosaic and NGRDI thresholding for different shore conditions: (**a**)—shallow sandy shores; (**b**)—steep banks covered with land vegetation.

**Figure 5 sensors-23-00303-f005:**
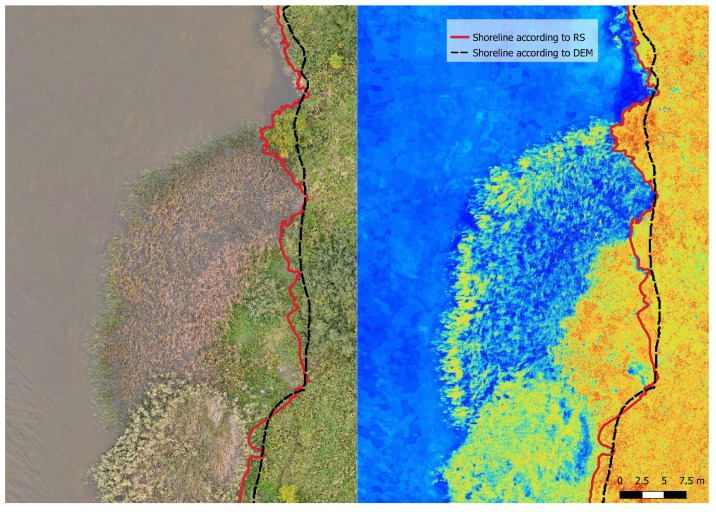
Shoreline detection in the areas containing medium-density aquatic vegetation.

**Figure 6 sensors-23-00303-f006:**
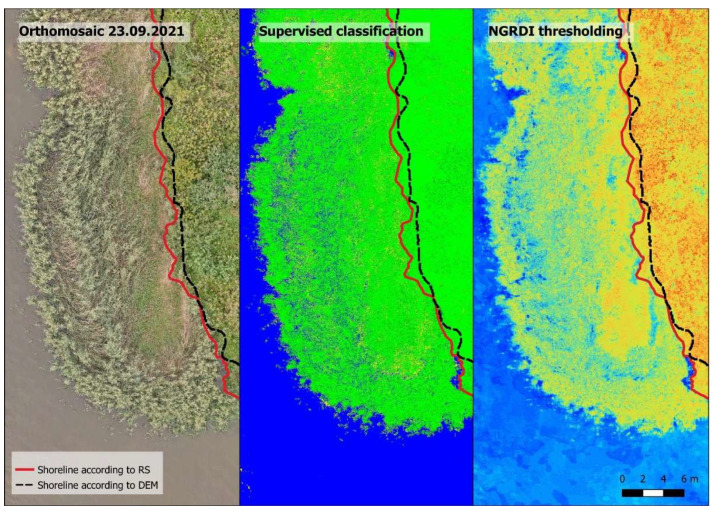
Shoreline detection in the areas containing high-density aquatic vegetation.

**Figure 7 sensors-23-00303-f007:**
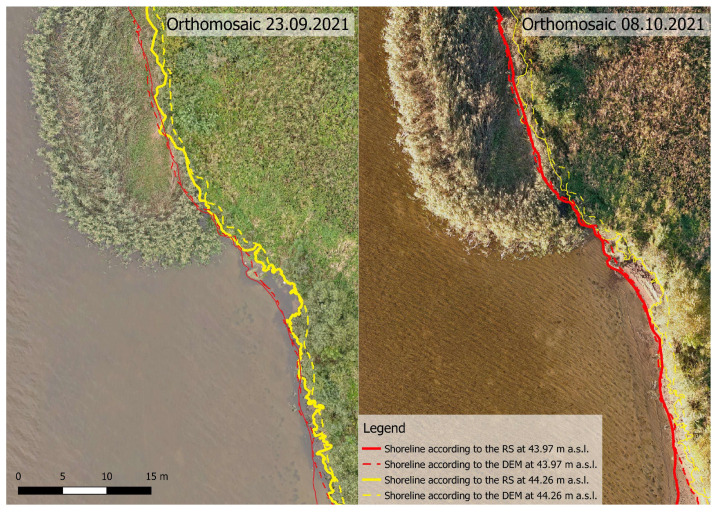
Shoreline detection comparison derived from traditional surveying and remote sensing.

**Figure 8 sensors-23-00303-f008:**
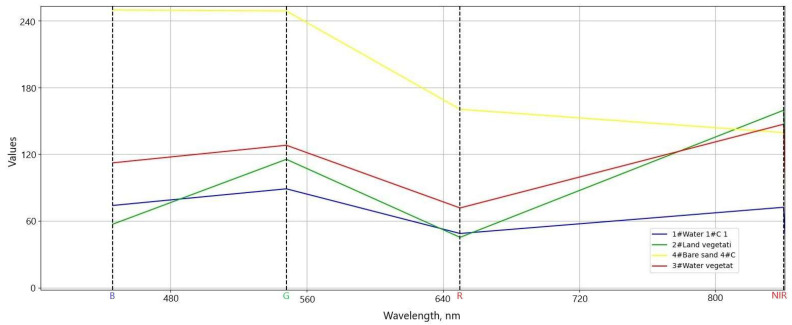
Spectral signatures of different surfaces: water, sand, land, and water vegetation.

**Table 1 sensors-23-00303-t001:** Performed flight missions with the UAVs.

Date	UAV Used	Water Level in the Reservoir, m a.s.l.
08.06.2021	DJI Martice 200	43.88
23.08.2021	Autel EVO II Pro	44.23
01.09.2021	Autel EVO II Pro	44.16
23.09.2021	Autel EVO II Pro	44.26
08.10.2021	Autel EVO II Pro	43.97

**Table 2 sensors-23-00303-t002:** Accuracy assessment of unsupervised index classification.

Evaluation Indices	Unsupervised Classification of			
	NDVI	NDWI	NGRDI	VARI
Overall Accuracy, %	93.4	97.4	92.0	87.4
Kappa Coefficient	0.867	0.947	0.837	0.744

**Table 3 sensors-23-00303-t003:** Statistical analysis of the spectral reflectance of land and water vegetation.

Wavelength, nm	446 (Blue)	548 (Green)	650 (Red)	840 (NIR)
Spectral signature of land vegetation				
Values	57.140	115.434	45.338	159.510
Standard deviation	24.382	28.731	19.218	32.831
Spectral signature of water vegetation				
Values	112.262	128.086	71.747	146.954
Standard deviation	50.831	47.013	37.714	26.790

## Data Availability

Not applicable.

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
