# Peer review of "Monitoring Dewatering Fish Spawning Sites in the Reservoir of a Large Hydropower Plant in a Lowland Country Using Unmanned Aerial Vehicles"

_sensors, 2022, doi:10.3390/s23010303_

Round 1

Reviewer 1 Report

The authors describe their efforts to define the shoreline of a water dam by remote sensing data. The classical methods of bathymetric measurements in situ are labor intensive, time consuming. Using newly emerging technologies like LiDAR, aerial photography by drones is their research approach for modernizing the process of drawing shore lines. Techniques such as spectral analysis of reflected electromagnetic waves are used to estimate remotely the presence of water at a given pixel in observed data.

The authors describe the evolution of their research efforts and the difficulties they have encountered. They show examples of easy-to-detect-shore-line cases and more difficult ones. Difficulties persists when trying to identify the shore line at spawning sites where water vegetation can be dance and obscures the water.

The results of this piece of research can be used to identify all spawning sites in the water dam and other areas vulnerable to the water level fluctuations resulting from the use of water for power generation. This would help to bring up-to-date the instructions for water use of the most significant water dam in Lithuania for the benefit of all stakeholders – the authors say in conclusion.

The manuscript is well written. Language errors are a rarity. In line 436: „... methods have an clear advantage ...“ „AN“ should probably be „A“.

Author Response

Thank you kindly for the review. Please see the attachment for our responses.

Author Response

(The authors gave the same response as above.)

Reviewer 3 Report

Reviewer comments

Thank you for submitting your paper to Journal of Sensors. I read carefully manuscript number: sensors-2075284, the manuscript entitled: "Monitoring dewatering fish spawning sites in the reservoir of a large hydropower plant in a lowland country using remote surveying methods". This paper presents research concerning dewatered areas in the littoral zones of the Kaunas hydropower plant (HPP) reservoir in Lithuania. In my point of view, the result of this kind of research could be interesting and useful for many applications. Please check the English grammar. The English language is moderate. Please check all parts of the manuscript and correct grammatical errors. Some sections of paper require major revisions before any further. I attached my reviewer supplementary comments in the pdf file.

Author Response

(The authors gave the same response as above.)

Round 2

Reviewer 2 Report

Thank you for your efforts and I am satisfied with your replay.

Reviewer 3 Report

Thank you for submitting your revised paper to Journal of Sensors. I read the revised manuscript number: sensors- 2075284, the manuscript entitled: " Monitoring dewatering fish spawning sites in the reservoir of a large hydropower plant in a lowland country using remote surveying methods". The authors applied all comments point by point and I confirm their revision. The added information is important and useful and led to improving the manuscript. I accept the revised manuscript in this present form. I concur; the final decision is accepted for publication.